# Development of a New KPI for the Economic Quantification of Six Big Losses and Its Implementation in a Cyber Physical System

**Paula Morella [1],\***, **María Pilar Lambán [1]**, **Jesús Royo [1]**, **Juan Carlos Sánchez [2]** and **Jaime Latapia [2]**

[1] Design and Manufacturing Engineering Department, Universidad de Zaragoza, 50018 Zaragoza, Spain; plamban@unizar.es (M.P.L.); jaroyo@unizar.es (J.R.)

[2] Smart Systems, Tecnalia, 20009 Donostia-San Sebastian, Spain; jcarlos.sanchez@tecnalia.com (J.C.S.); jaime.latapia@tecnalia.com (J.L.)

\* Correspondence: 620453@unizar.es

**Abstract:** The purpose of this work is to develop a new Key Performance Indicator (KPI) that can quantify the cost of Six Big Losses developed by Nakajima and implements it in a Cyber Physical System (CPS), achieving a real-time monitorization of the KPI. This paper follows the methodology explained below. A cost model has been used to accurately develop this indicator together with the Six Big Losses description. At the same time, the machine tool has been integrated into a CPS, enhancing the real-time data acquisition, using the Industry 4.0 technologies. Once the KPI has been defined, we have developed the software that can turn these real-time data into relevant information (using Python) through the calculation of our indicator. Finally, we have carried out a case of study showing our new KPI results and comparing them to other indicators related with the Six Big Losses but in different dimensions. As a result, our research quantifies economically the Six Big Losses, enhances the detection of the bigger ones to improve them, and enlightens the importance of paying attention to different dimensions, mainly, the productive, sustainable, and economic at the same time.

**Keywords:** Cyber Physical Systems; Key Performance Indicator; Industry 4.0; cost model

## 1. Introduction

Industry 4.0 represents an industrial production transformation through the merging of the Internet and information and communication technologies (ICT) with traditional manufacturing processes [1]. The fourth industrial revolution, known as Industry 4.0, was born in Hannover in 2011, and has improved the production systems of European companies [2]. Digital technologies in Industry 4.0 enable the improvement of manufacturing, economic, and environmental performance through information gathering and processing [3]. This information sharing along the supply chain (SC) allows, among other things, real-time integration of supply chain partners, and improves production planning, reducing environmental uncertainty and improving SC efficiency [4]. Furthermore, Industry 4.0 and digitization contribute to the quick and flexible response of companies to market changes [5].

A new generation of smart factories is emerging due to this fourth industrial revolution and its technologies [6], such as the Cyber Physical Systems (CPS), smart systems that encompass computational and physical components [7]. CPS enables us to acquire real-time data for making decisions [8]. Internet of Things (IoT) technology enables the communication between physical resources and together with CPS can capture and communicate data accurately and consistently [2]. This real-time process enhances the idea to create an accurate costing system based on it and the development and calculation of real-time indicators. However, this idea has not been further developed [9].

Moreover, the implementation of Industry 4.0 technologies can involve cost reduction directly, i.e., labor costs, or can be implemented for cost monitorization, showing how it can be reduced, which is part of the aim of this research. Prajogo and Olhager [10] suggested that technology-enabled SC platforms can contribute to a decrease in production costs. Nguyen et al. [11] asserted that, thanks to an efficient information processing, digital technologies enhance the decisions of production planning and control, which increase operational efficiency and profits, and reduce the cost. Particularly talking about CPS, this decrease in cost can also be seen. Matana et al. [12] said that the incorporation of CPS technologies into internal logistic equipment can reduce logistics operational costs and production cycle times. PwC considered the reduction of manufacturing costs as one of the benefits of CPS. Furthermore, they consider reductions of up to 13.8% in manufacturing costs and 20% revenue increase by 2020.

Particularly, our research is focused on the development and implementation in CPS of indicators related to the six big losses of Nakajima, one of the most important Lean Manufacturing techniques. Lean methodology framework is being an interesting topic for many researchers in recent years [13] and is one of the most common strategies to achieve better results and higher competitiveness [14]. Lean Manufacturing techniques are based on continuous improvement and removal of wastes, such as time, resources, or economic wastes [15]. Their motto "less is more" has encouraged companies to enhance the product and process value [16]. This methodology includes a set of operational tools that enhance the identification and reduction of wastes and non-value added activities, reducing costs, and improving efficiency and effectiveness of production processes [17]. As can be seen, Lean Manufacturing methodology is directly related to cost reduction. Machado et al. [18] asserted that the implementation of lean manufacturing tools is beneficial for companies, thus, they can obtain advantages in quality, inventory management, elimination of losses, and improvement of the financial and operational controls. Particularly, this identification and reduction of losses, which are considered as activities that do not add value, but add cost to the product that customers are not willing to pay, allows the reduction in manufacturing costs [19].

The three Mus, Muda, Mura, and Muri, are some of the techniques to detect wastes. Defects, overproduction, waiting, transportation, inventory, motion, and over-processing are seven types of wastes covered by Muda, while Muri refers to any action related to tangible or intangible stress conditions, and Mura identifies irregular machine or person use [20]. Regarding losses, the six big losses concept developed by Nakajima is currently used in production processes that involve machining processes, and it has been chosen to establish our new KPI based on cost losses. These losses were categorized by Nakajima into three big blocks, i.e., time, speed, and quality loss, described as follows (see Figure 1) [21]:

- Breakdowns or equipment failures: time losses.
- Set-up and adjustments.
- Minor stoppages during production.
- Reduced speed.
- Reworks: products with minor defects that can be reworked.
- Rejects: defective products that cannot be reworked.

These six big losses were used by Nakajima to develop an indicator, known as Overall Equipment Effectiveness (OEE). This OEE measures the productivity of individual equipment in a manufacturing environment. It is calculated by multiplying three indicators: availability, performance, and quality [22].

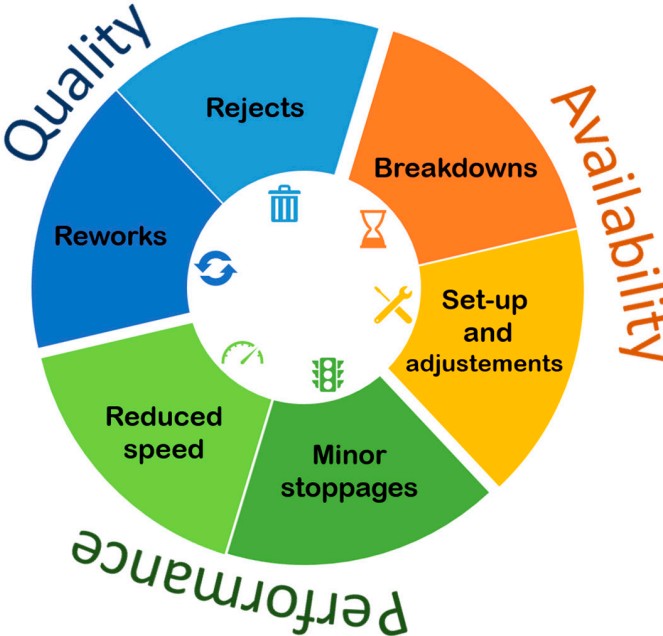

**Figure 1.** Six big losses.

Although these three indicators are key elements of performance, other relevant factors, e.g., efficient use of raw materials, their quality, or costs are not taken into account in the OEE calculation [23]. Over the years, the literature scarcely considered any references to the economic aspects associated with the impact of the losses indicated by Nakajima [24]. Although some criticisms are related to its non-connection with costs [25], Muchiri and Pintelon [22] considered that the translation of loss effectiveness in terms of cost should be explored and Parida and Kumar [26] considered cost measure as part of the internal effectiveness for their total maintenance effectiveness calculation.

These six big losses had been used to develop other KPIs, e.g., ECL (Energy Consumption Loses), which takes part of our previous work and shows the energy losses that are involved in a machining process [27]. In order to compare our new KPI with them, both KPIs, OEE, and ECL, have been implemented in our CPS.

KPIs and manufacturing targets, such as production cost, are indispensable for efficient operation of a manufacturing enterprise. Therefore, the design, production, and manufacturing decision variables need to be identified, modeled and compared against them [28]. Baykasoglu and Kaplanoglu [29] remarked how a company that wants to maintain a competitive position must be able to combine high quality product and/or services with fine-tuned delivery times and the lowest possible costs. In order to minimize costs, exact information about those costs must be available [30].

Regarding cost, the identification of KPIs, which allows enterprises to control and reduce costs, is crucial. According to Needy et al. [31] the selection, evaluation and configuration of systems should be based on economic decisions. Brüggemann and Bremer [32] talked about the disadvantage of having excessive production costs in markets characterized by predatory competition. Cost can be defined in terms of financial outlay or as an effort made to achieve an objective [33], and its plan, control, prediction, and reduction in operating processes allow companies to achieve revenue streams and to assess the financial implications [34].

Industry 4.0 has remodeled the data acquisition in enterprises. Not only allow the development of CPS enterprises to collect real-time data, but they can develop KPIs in real-time too. In comparison to traditional accounting systems, Industry 4.0 systems are able to measure all the needed metrics and to report them faster. These systems can manage value streams, identify wastes, make each employee accountable for cost reductions, and link all reporting to improvement cycles [9]. Although real-time has not received in recent years much attention in the literature [35], real-time information

about machines enhance the creation of a system based on actual time requirements, and it can be observed the development of profitability over time detecting changes earlier [9]. In addition, having information in real time is a very important aspect of adequate decision-making in the business world.

There is no needed to develop complex cost indicators, a good cost model development could provide relevant information and most of the time is the base to develop a cost indicator. According to Dogan and Aydin [36] several cost modelling strategies have been developed to characterize production cost, i.e., activity-based costing, product-based costing, process-based costing, bottom-up costing, and top-down costing. Panicker et al. [28] modelled a production cost based on six cost components: facility cost, capital cost, utilities costs, raw material cost, labor costs, and maintenance costs which are dependent upon factors such as manufacturing location, type of manufacturing process, raw materials used, source of raw materials, and transportation modes. His research estimated the most favorable manufacturing decisions to achieve a specific KPI value translating the cost model into a Bayesian Network (BN). Other authors have developed some cost KPIs. Gunasekaran et al. [37] described a Performance-Based Costing (PBC) system, which focuses on performance, no activities. It avoids product cost information that can have been distorted by traditional cost systems in virtual SC. PBC identifies business areas that are able to add value to an organization and calculates direct materials, labor or overhead to accurately estimate production cost. Christen et al. [38] presented a Cost Performance Indicator (CPI) curve. This curve plotted a cost variable against one characteristic, which is involved in cost variable, thus, the cost evolution through this characteristic is shown. Windmark et al. [39] implemented a Cost Performance Ratio (CPR), which allowed to describe a cost (for a cutting tool, a workpiece material, machine units . . . ) with respect to their performance and user value. Otherwise, Sepherd et al. [40] collected a huge number of cost and non-cost indicators, which can be used in an SC. Focusing on cost indicators, cost of goods sold, value added productivity, percentage sales of new product compared with wholesales for a period, work in process and cost per operating hour, are some of them. Finally, Wudhikarn et al. [41] developed a cost KPI (Overall Equipment Cost Loss—OECL) based on the three major elements used in the OEE approach but represented in monetary units. His proposal considers factors to obtain the economic losses associated with each of the variables of the OEE. The number of pieces manufactured of a specific reference or product is related to the number of total pieces manufactured [42], assuming that all the manufactures are equal and that they need the same time to be manufactured. However, the unitary times of manufacture can vary depending on the characteristics of the geometry and the materials, among others. Therefore, his proposal does not allow to contemplate the temporal variability between the manufacture of different pieces when calculating the losses contemplated by Nakajima, and between different references. He considers that the time to manufacture the pieces produced in a resource is analogous and uses the number of pieces manufactured to obtain the cost distribution of a specific reference. As in the ABC (Activity-based costing) systems, which assist managers in important strategic business decision-making [43] and use a large number of cost drivers to count the number of times an activity was performed, including the number of production cycles, in the production environment [44]. When the resources that are necessary for the development of the activity vary, that is, when it does not cost the same to make different assemblies, or when some geometries are more complicated, the transaction drivers do not bring precision to the calculation of OECL. Given the importance of time, a new system, TDABC (Time-Driven Activity-Based Costing) was developed. This system uses temporal equations, which are more accurate, sharing the global activity costs over specific products [45], but they are also more expensive and complex to measure. Industry 4.0 seems to be the solution to this weakness because it allows values to be obtained in real time.

It can be deduced that working in real-time has not received much attention yet, even though Industry 4.0 enhances its use, e.g., using CPS for real-time data acquisition. The use of this CPS allows the acquisition of time variables, which are more accurate than the average times, that are often used as an approximation. Regarding to the six big losses research, it has been further developed in terms of productivity, but not in other dimensions, such as the sustainable or economic ones. Therefore,

we have previously researched the sustainable dimension of the six big losses, as can be seen in [27], and now the objective of this study is to fill those gaps developing a cost model and a new cost KPI capable of measuring in real-time the impact of cost in the six big losses developed by Nakajima. Both the cost model and the KPI are implemented in a CPS, which allows us their real-time calculation. Not only does this new KPI enhance the decision-making, but it also shows the most effective points to decrease production costs. Moreover, it completes the dimensions in which the six big losses can be represented, representing them in an economic dimension and allowing companies to develop a multidisciplinary decision-making tool, comparing the six big losses in a productive, sustainable and economic dimension. After the literature review of the key research concepts and their correlation. This manuscript goes ahead with the following research methodology. A cost model has been analyzed and adapted to our research for developing our new KPI and its equations. After that, it has been implemented in a CPS. On that note, a case of study has been carried out to show this KPIs development and implementation. The results have been discussed and compared with other KPIs. Conclusions and proposals for future studies are established at the end of the paper.

## 2. Materials and Methods

Our research is based on two objectives: the development of a cost KPI that is able to monetary quantify the six big losses of Nakajima and the implementation and use of CPS to calculate this KPI in real time.

Our KPI development is based on the cost model presented by Lambán [46], which has been studied and modified to obtain a cost model aligned with the CPS acquisition system. Once the cost model was developed, we followed a similar philosophy as Wudhikarn et al. [41] used to develop OECL, because his research is based on the OEE major elements, which come from the six big losses, which are also our starting point. However, our research goes deeper into the economic evaluation of losses, presents a cost model and uses real time data, pretending to be the third element which is needed to evaluate the six big losses in three fundamental dimensions: productive (OEE), environmental (ECL) and economic (our new KPI). Whereas OECL pretends to evaluate several pieces of equipment and to rank machines [47] and use the number of pieces to calculate the indicator, which is less accurate than our method based on real time data acquisition using CPS, as explained in Introduction.

### 2.1. Modelling Cost

The cost model presented by Lambán [46] has been modified, focusing on direct costs, which are the most significance for our KPI development. This cost model considers the raw material and the different phases which are included in a product machining process, from its design to its manufacture. Below is an outline of our cost model:

1. Raw Material: the sum of the cost of acquisition, supply, transport, storage, inspection and internal transport of raw material.
2. Process of manufacturing operational: divided in previous and operational phases.

   a. Previous phases:

      i. Part design: labor cost and equipment cost involved in this process.
      ii. Machine programming: labor cost and equipment cost involved in this process.
      iii. Worksheet preparation: labor cost and equipment cost involved in this process.

   b. Operational phases:

      i. Direct labor: labor costs involved in operational processing.
      ii. Machine:

         1. Transformed resource: it could be owned or rented.
         2. Consumables: the sum of cost of tools, tooling, energy and fluids.

The total sum of all cost model components gives as a result the cost per piece manufactured. However, we use the itemized cost model for our new KPI implementation, as can be seen in Table 1.

**Table 1.** Database parameters.

| Parameter | Value | Observations |
|---|---|---|
| Labor cost | 12 €/h | One operator per action |
| Informatics resources | 0 €/h | There is no maintenance or licenses |
| Transform resource | 5.84 €/h | 8 years of amortization |
| Tools | 0.34 €/h | 15 min of useful life |
| Tooling | 0.55 €/h | Only 1 tooling is used |
| Fluids | 0.21 €/h | Distilled water and coolant |
| Energy | 0.135 €/kWh | Standard electric rate |
| Preparation time | 1 h | Measured in the workshop |
| Programming time | 1 h | Measured in the workshop |
| Number of rejects | 1 | Hypothesis |
| Number of reworks | 1 | Hypothesis |

*2.2. Application of the Six Big Losses to Cost*

In the same way that we related in [27] the six big losses with the energy consumption, the six big losses are applied to the economic dimension as follows:

1. The six big losses, traditionally referred to time losses, have been adapted to cost as follows: Breakdown and adjustment losses: all costs associated with maintenance and adjustment actions, the opportunity cost loosed and labor and machine costs involved in this process.
2. Waiting time losses: opportunity cost loosed, and labor and machine costs associated with set-up and adjustments, e.g., time during tool or reference changes.
3. Reduced speed losses: opportunity cost loosed, and costs associated with the difference of time between the machine possible working speed and its real one.
4. Minor stoppage losses: short interruptions by process jams, incorrect feed, flow obstruction, checks, or stops leading to additional flow time. In this case, the opportunity cost loosed, and stops of maximum five minutes, although this maximum must be established for each company.
5. Rework losses: costs of rework activity when the product has minor defects.
6. Reject losses: opportunity cost loosed, and labor and machine costs generated by rejecting products.

*2.3. Cost KPI Definition: Cost Loss Indicator (CLI)*

Our new KPI (CLI) is defined as the costs associated with the six big losses during a specific period, e.g., a working day. Each component of the KPI is composed by different costs which can be seen in Figure 2. The main cost components are opportunity cost (OC), referring to benefit loss, and production cost (PC).

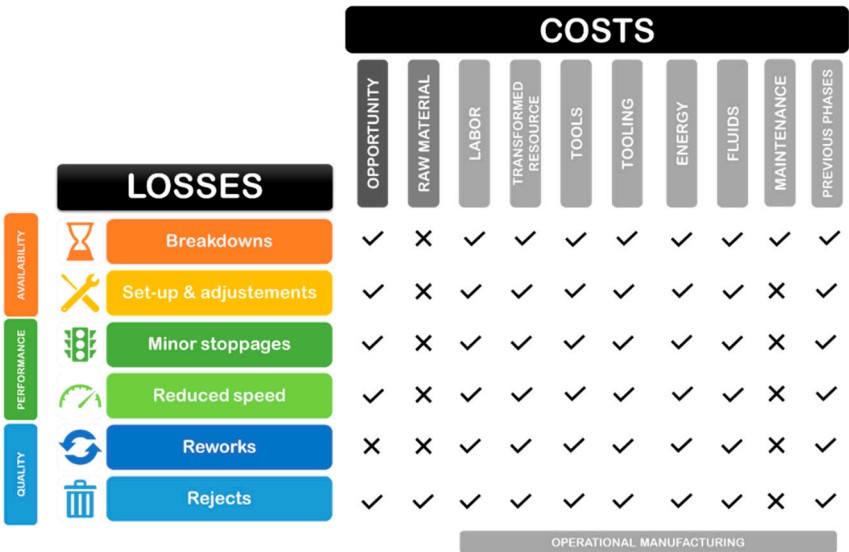

**Figure 2.** Costs for each loss.

This KPI is obtained by adding three components:

$$CLI = C_{AL} + C_{PL} + C_{QL} \tag{1}$$

where $C_{AL}$, $C_{PL}$, and $C_{QL}$ are the availability, performance, and quality cost losses, respectively.

$C_{AL}$ can be expressed as:

$$C_{AL} = OC_A + PC_A \tag{2}$$

$$OC_A = \sum_{j=0}^{m} \left( \frac{n_j}{\sum_{i=0}^{n} \text{Machining time of i}} [u/s] \times t_{AL} [s] \times f_j \times Bj [€/u] \right) \tag{3}$$

$$PC_A = \sum_{j=0}^{m} \left( t_{AL} [h] \times f_j \times \text{Availability Cost j} [€/h] \right) \tag{4}$$

where $t_{AL}$ is the time corresponding to breakdowns, adjustments and waits (the two big losses related to availability), m is each machining reference, n the total number of machining parts per reference, B is the benefit calculated as the difference between its sale price and its production cost, Availability Cost is the sum of costs reflected in Table 1, and $f_j$ is defined as follow:

$$f_j = \frac{\sum_{j=0}^{m} \text{total time reference j}}{\text{total time}} \tag{5}$$

Being total time the period in which the Six Big Losses are calculated, e.g., a working day.

The $C_{PL}$ is expressed as:

$$C_{PL} = OC_P + PC_P \tag{6}$$

$$OC_P = \sum_{j=0}^{m} \left( \frac{n_j}{\sum_{i=0}^{n} \text{Machining time of i}} [u/s] \times t_{PL} [s] \times f_j \times Bj [€/u] \right) \tag{7}$$

$$PC_P = \sum_{j=0}^{m} \left( t_{PL} [h] \times f_j \times \text{Performance Cost j} [€/h] \right) \tag{8}$$

where $t_{PL}$ is the performance losses time, associated with reduced speed and minor stoppage losses, Performance Cost is the sum of costs reflected in Table 1 and the rest of parameters has been defined before.

The $C_{QL}$ is composed by rejects and reworks:

$$C_{QL} = C_{QL_{rej}} + C_{QL_{rew}} \tag{9}$$

The $C_{QL_{rej}}$ adds a new component, the material cost (MC):

$$C_{QL_{rej}} = OC_{Qrej} + PC_{Qrej} + MC_{rej} \tag{10}$$

$$OC_{Qrej} = \sum_{j=0}^{m} (p[u] \times \text{Benefit j } [\text{€/u}]) \tag{11}$$

$$PC_{Qrej} = \sum_{j=0}^{m} \left[ \left( \sum_{k=0}^{p} \text{machining time of k } [h] \right) \times \text{Reject Cost i } [\text{€/h}] \right] \tag{12}$$

$$MC_{Qrej} = \sum_{j=0}^{m} (p[u] \times \text{Material Cost j } [\text{€/u}]) \tag{13}$$

where p is the number of rejects for each reference. All these parameters have been defined before, except Rework Cost (RC), which is defined in Table 1.

The $C_{QL_{rew}}$ is expressed as:

$$C_{QL_{rew}} = RC \tag{14}$$

$$RC = \sum_{j=0}^{m} \left( \left( \sum_{k=0}^{q} \text{machining time of k } [h] \right) \times \text{Rework Cost m } [\text{€/h}] \right) \tag{15}$$

where q is the number of reworks for each reference.

## 2.4. CPS Implementation

Our second objective consists in showing the contribution of Industry 4.0 technologies, especially CPS, to improve manufacturing systems by implementing a CPS in a machine tool, which allows us to develop an automatic real-time calculus of KPIs. The combination of Industry 4.0 technologies allows the acquisition of an accurate collection of real-time data [48].

CPS enhances the integration between real and virtual world [49], which is crucial for monitorization and control [50]. There are several examples of CPS used for monitoring, real-time data acquisition and KPIs implementation. Mörth et al. [51] developed a case study of a CPS demonstrator for performance monitoring in intralogistics, from the acquisition and processing of data to the visualization of performance information on a dashboard. Tan et al. [52] studied the real-time data acquisition to model, plan and schedule the industrial robot assembly process. Gürdür et al. [53] monitored KPIs to control operations and decisions in automated warehouses, as a form of CPS.

Our CPS is located in the Design and Manufacturing Engineering Department of the University of Zaragoza, where we are creating a Lab 4.0 in collaboration with Tecnalia. It is composed by a machine tool, a monitoring machinery system and an industrial computer. Our machine tool (HAAS VF-3) is a five-axis vertical milling machine. The system of monitoring machinery was added to this machine tool. Both were then connected to the industrial computer (Beckhoff), which can capture data in real time and store it in the cloud. Thus, the machine tool turned into a CPS (see Figure 3). Once the CPS is implemented the acquired data are treated and used for KPI calculations by Python. This programming language has been used due to its capability to develop software quickly and easily [54].

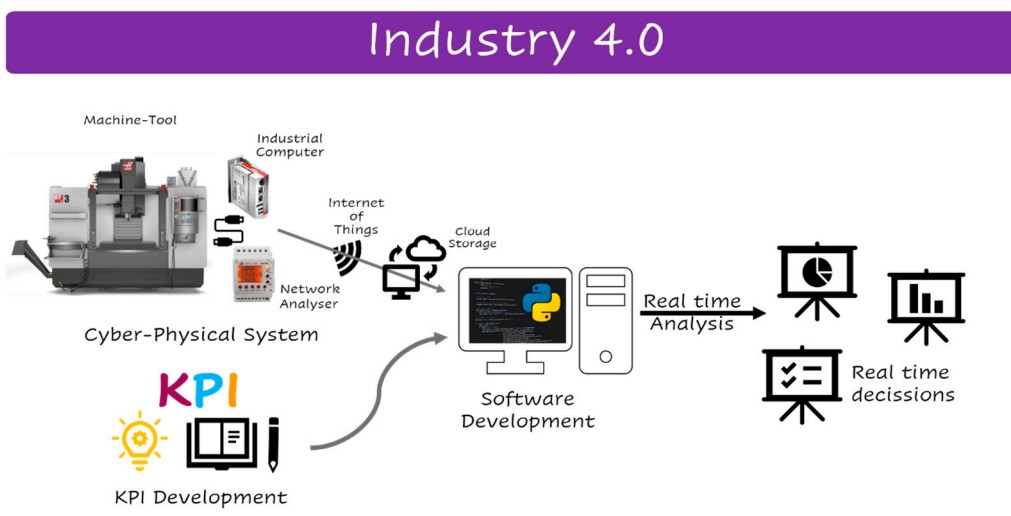

**Figure 3.** Cyber Physical System (CPS) implementation to data acquisition.

### 2.5. Calculating Cost KPI in a CPS

Once the CPS is implemented, the most appropriate variables for KPI calculation in real-time are analyzed, checked, and selected. Python has been used to develop the software that translates the acquired data into relevant information and indicators. Our system can capture variables, such as machine time, energy consumption, part number, tool number, tool changes and the spindle rotation. However, other parameters, which must be given, are stored in a database and can be seen in Table 1. Then, the software can calculate KPIs, such as OEE or ECL, which are defined in [27], our model cost defined in Section 2.1 and our new KPI defined in Section 2.3 using the acquired data and the established parameters. For example, to calculate the energy cost of six big losses, the CPS can capture the energy consumption associated with each one of the six big losses. As the CPS can determine the process that the system is carried out (how long it takes, how much energy the machine consumes, if the machine is working properly or not, and if the machine is working at reduced speed). Once the CPS knows this energy consumption, it can obtain from the database the energy cost per kWh. Figure 4, which is an extension of the flow chart developed in [27], represents a flow chart for the KPIs implementation in a CPS, these KPIs will be compared later (OEE, ECL and ECI).

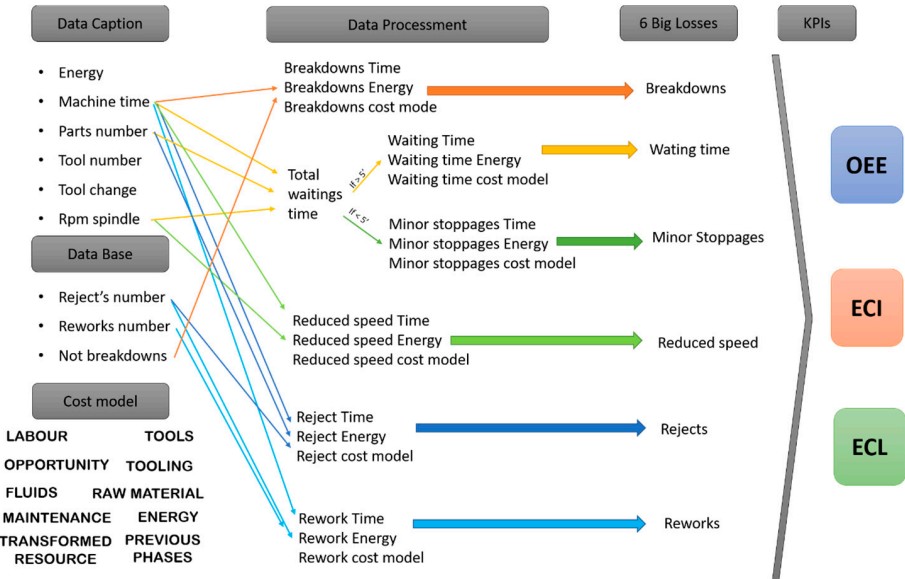

**Figure 4.** Flow chart for Key Performance Indicators (KPIs) implementation in CPS. Reference [27].

The difficulty to distinguish the durations associated with each of the six big losses in a conventional machine poses a problem for these KPIs calculations on them. However, the CPS implementation enhances this performance, due to its ability to know what type of activity is occurring in real time.

## 3. Results and Discussion

This section carries out a case of study and presents and analyzes the results obtained from the case of study. This case of study is the same that was developed in our previous research [27], thus we can compare these results with our previous ones. The results are presented as follows: the results of ECI are shown, and then its results are compared with ECL and OEE. Thereby we are able to compare the six big losses through three different and relevant dimensions: effectiveness, sustainability and economy, thus, we avoid the misleading picture of machining performance that can produce the exclusive rely on cost indicators [40].

### 3.1. Test Description

In Figure 5 can be seen the piece which has been designed to perform this case of study. It has been machined from aluminum blocks, whose dimensions were 100 mm × 100 mm × 20 mm, and the operations (OP) below were performed on it:

- OP1: face milling on the top side with an octagonal plate of Ø80 of five plates.
- OP2: face milling at 90 degrees with a plate of Ø63 with five plates.
- OP3: slot milling with two plates with a cutter of Ø25.
- OP4: shoulder milling with cutter of Ø14 of three cuts and a cut to the center; in other words, the tool has an edge toward the center and low cutting.
- OP5: circular slot milling with a fast steel cutter of Ø10 HSS (High Speed Steel) with three cuts and a center cut.

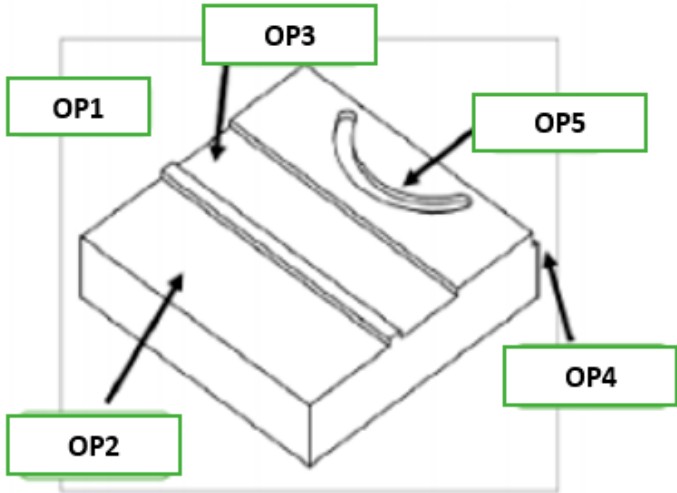

**Figure 5.** Machined part design during testing.

Our case of study consists of three cases, in each case five equal pieces are manufactured at different tool feed rates, case 1 at 100 %, case 2 at 75 % and case 3 at 50 % of the feed rate that the manufacturer recommends according to the material and quality selection. Each case has a different duration, i.e., case 1:23 min, case 2:26 min, and case 3:33 min.

As hypotheses, it has been assumed that there had been one reject and one rework, for each case, and that there had been no breakdowns.

Our CPS collected the real-time data from each case and processed them with three programs, to obtain the ECI, the ECL and the OEE of each case.

### 3.2. CLI Results

First of all, the cost model, which has been developed by our own (see Section 2.1), enables us to obtain the total unitary production cost (see Table 2) and every term of it (see Table 1). The total unit production cost is used to calculate the benefit obtained for each reference as the sale price minus the production cost.

**Table 2.** Unit production costs.

|  | Case 1 | Case 2 | Case 3 |
|---|---|---|---|
| Unit production cost | 11.84 € | 12.13 € | 12.71 € |

As a result of the formulas implementation (see Section 2.3) to obtain the new KPI and the program's development in Python, the new KPI values are obtained on availability, performance and quality components, which can be seen in Table 3. Comparing Tables 2 and 3, can be concluded that the bigger the production cost is, the bigger the CLI is too. A graphical comparison of each CLI case can be seen in Figure 6.

**Table 3.** Test results.

|  | Case 1 | Case 2 | Case 3 |
|---|---|---|---|
| Section | 2.53 € | 2.51 € | 2.40 € |
| Section | 3.70 € | 4.67 € | 5.68 € |
| Section | 10.90 € | 10.67 € | 10.27 € |
| Section | 17.13 € | 17.85 € | 18.35 € |

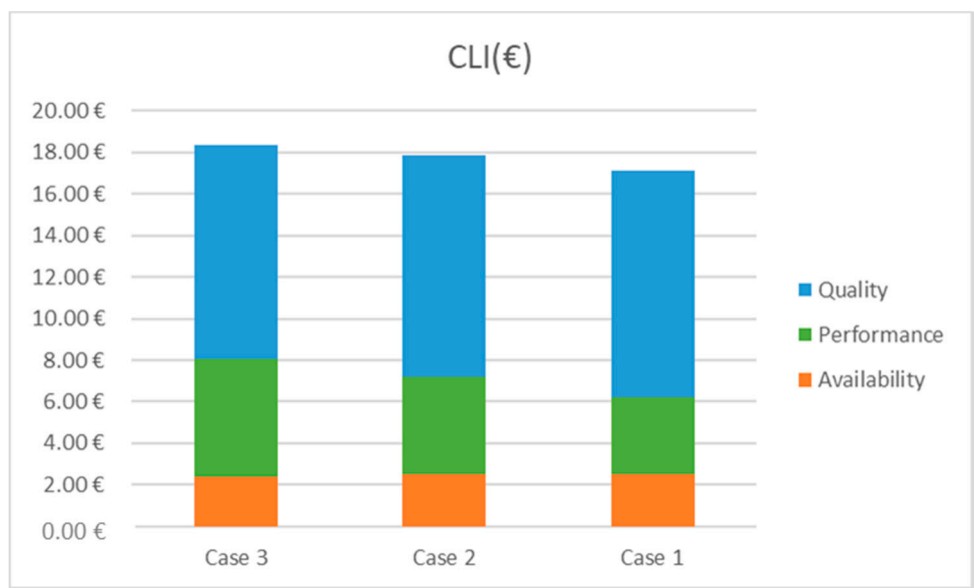

**Figure 6.** Stacked bar chart for the three tests.

Observing these results, it can be said that case 1 reduces cost losses by 3% and 7% in comparison to machining in cases 2 and 3.

Although the highest loss is caused by quality, the biggest difference between each case is associated with performance due to the difference in reduced speed losses. Quality losses are quite similar because the number of rejects and reworks has been a hypothesis which is the same in the three cases, the difference in this cost is defined by the machining part time on each case. Availability is the lowest loss because there were no breakdowns during the study.

Figures 7–9 are Pareto diagrams, which improve the analysis of each case, allowing the identification of 20% of the causes that lead to 80% of the effects.

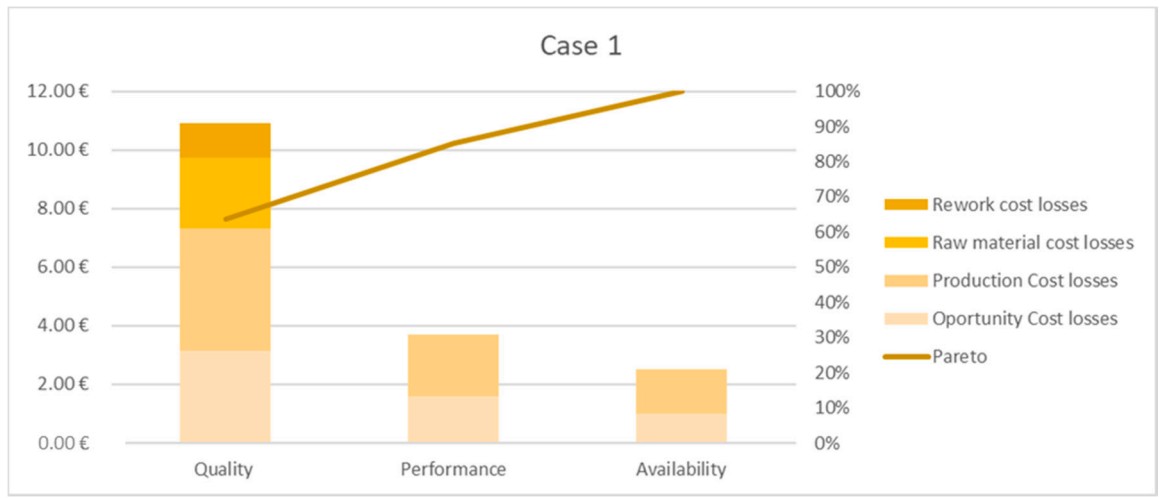

**Figure 7.** Cost loss indicator (CLI) case 1 diagram.

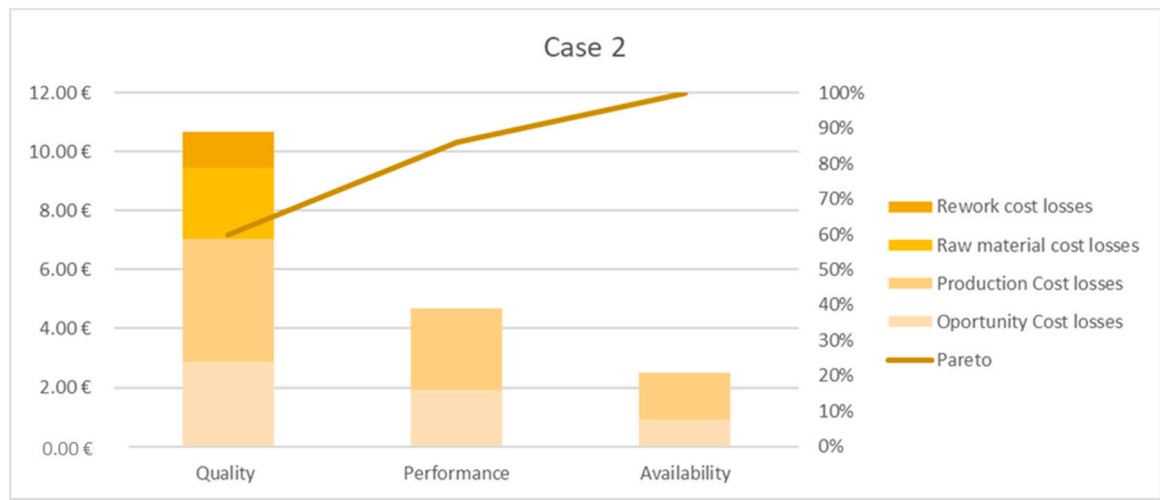

**Figure 8.** CLI case 2 diagram.

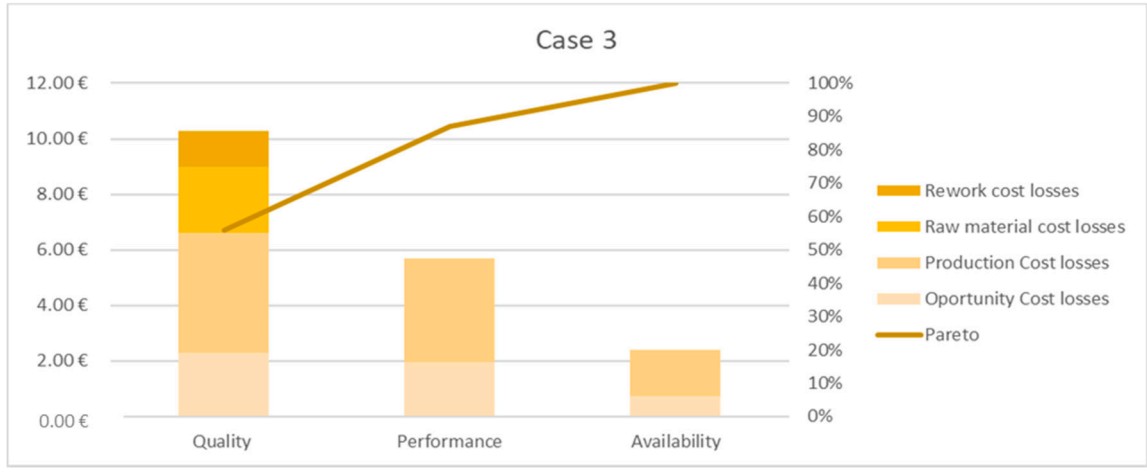

**Figure 9.** CLI case 3 diagram.

The biggest losses can be identified from these diagrams. In the three tests quality losses involve more than 50% of the global losses, thus, should be the ones to be corrected first. The next problems to be solved are performance losses, in case 2 and 3 they are bigger due to their reduced speed. Furthermore, in the three cases, minimizing the first two losses involve almost 90% of the cost losses. Regarding to quality losses, rejects implies the biggest cost, therefore, rejects reduction should be the first measure to decrease these cost losses.

This analysis enhances the results' comprehension and highlights improvement and cost reduction areas for the company.

### 3.3. Comparison between New KPI, OEE, and OECL

The development of this new KPI and its implementation together with the OEE and ECL implementation, whose origin is the same: the six big losses, allow us to compare these three KPIs and therefore, to compare the productive, sustainable, and economic dimensions of the six big losses (see Table 4 and Figure 10).

**Table 4.** Overall Equipment Effectiveness (OEE), Equipment Cost Loss (ECL), and new KPI results summary.

|        | OEE    | ECL (g $CO_2$ eq) | OECL (€) |
|--------|--------|-------------------|----------|
| Case 1 | 42.66% | 63.39             | 17.13 €  |
| Case 2 | 42.95% | 64.64             | 17.85 €  |
| Case 3 | 41.01% | 98.63             | 18.35 €  |

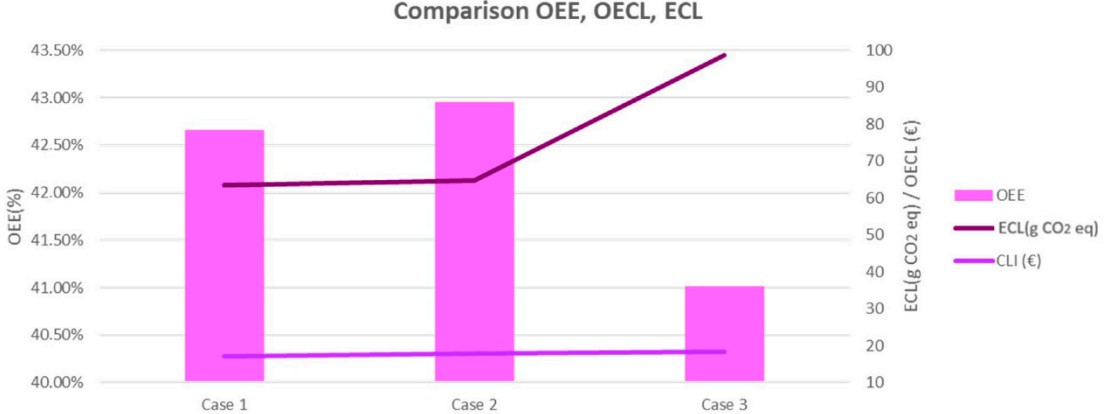

**Figure 10.** Comparison between OEE, ECL, and new KPI.

The comparison between OEE, ECL, and the new KPI shows that the highest percentage of OEE (case 2) does not correspond with the lowest ECL and new KPI. Whereas, case 1 has the lowest ECL and ECI and a lower OEE percentage. For that reason, having a larger OEE is not enough to conclude that a case is better than the others. The more diverse information we have about a process, the better decisions we can make about it.

A new indicator, known as OEE′, has been defined to improve this comparison between loss expressions. OEE′ shows the machine's unproductivity, so is the opposite of OEE, and its calculation can be seen at Equation (16)

$$OEE' = 1 - OEE \qquad (16)$$

Figure 11 shows the comparison between three loss expressions, OEE′, ECL, and new KPI. This diagram supports the same idea that the three components do not have to be the lowest necessarily at the same case.

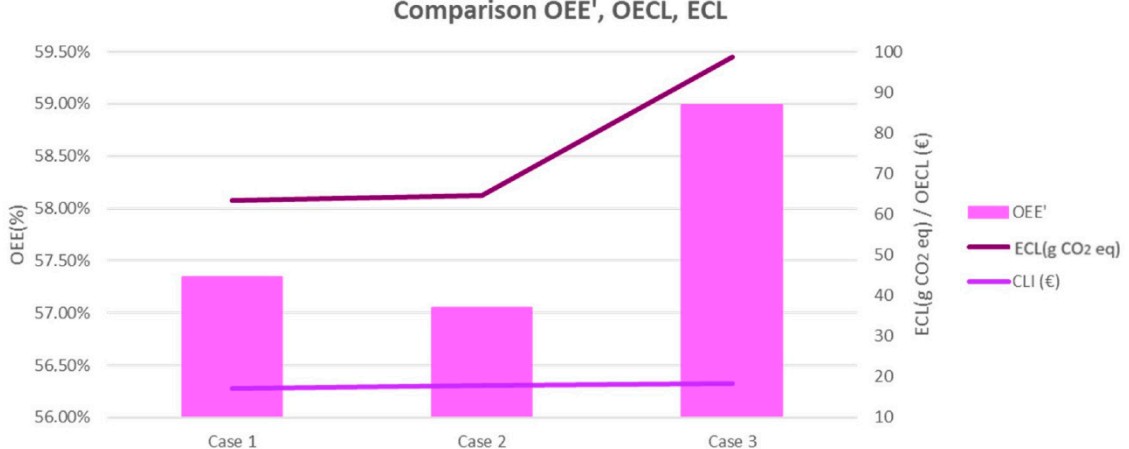

**Figure 11.** Comparison between OEE', ECL, and new KPI.

Our new cost KPI together with OEE and ECL provide companies with a stability between production, sustainability, and economy in machining processes that was not possible before our KPI development.

Moreover, the implementation of these KPIs in a CPS allows their calculation in real time and therefore, a machine tool monitorization that can be seen as a predictive maintenance tool. A sharp increase of these KPIs could be associated with a machine failure, thus, it should be checked.

## 4. Conclusions

The main objective of this project is the development of a new KPI that measures the six big losses in an economic dimension and can be implemented in a CPS to its real-time calculation. Moreover, it completes our previous research about six big losses in productive and sustainable dimensions.

Our research supports the idea that Industry 4.0 enhances the machining tool analysis and decision-making for companies. CPS is able to acquire real-time data, process them to obtain real-time indicators for machine tool monitorization and display several visual information through different diagrams and indicators which inform companies at a glance about the machine health. Moreover, the continuous monitorization has a predictive maintenance component, an abrupt and anormal value of one of these KPIs can be interpreted as a machine tool failure, thus, notice should be given for review.

The new KPI development and its comparison with the OEE, OEE', and ECL, whose origin is the same: the six big losses; it provides companies with a competitive advantage. Not only does it allow companies to compare the losses in three different areas: production, sustainability, and economy, but it also allows the development of a multidisciplinary decision-making tool. This multidisciplinary analysis is considered more reliable than single factor analysis, thus, production decisions should involve several parameters and variables [39].

This research has improved our knowledge about aspects such as the process of CPS implementation and its features, specially KPI implementation. With the acquired knowledge, we can extrapolate this to other SC processes and systems, which is one of the limitations of this research, given that this KPI implementation has been made for a single machine. For further research, the horizontally or vertically integration of these KPIs—towards other processes in the SC or towards other logistics operations- would be considered. The majority of the companies, which are using Industry 4.0 technologies or are nearly to use them can implement this technology in several resources, i.e., machining tools, 3D printers, Automated Guide Vehicles (AGVs), manufacturing systems, or forklifts. Furthermore, we have implemented a CPS in another machine-tool in order to create a machine park, expand the KPI development and analyze the communication machine to machine.

To conclude, the outcome of our research highlights the importance of attending to different dimensions (productive, sustainable, and economic) towards with the importance of implementing industry 4.0 technologies in companies to acquire a competitive advantage over the competition.

**Author Contributions:** Conceptualization, P.M. and M.P.L.; Investigation, P.M. and M.P.L.; Methodology, P.M. and M.P.L.; Project administration, J.R. and J.C.S.; Resources, J.L.; Software, P.M.; Supervision, M.P.L. and J.R.; Validation, M.P.L.; Writing—original draft, P.M.; Writing—review & editing, M.P.L. All authors have read and agreed to the published version of the manuscript.

**Funding:** This research received no external funding.

**Acknowledgments:** The authors would like to acknowledge the participation of the Precision Mechanic Service of the Universidad de Zaragoza during the essays, as well as, the collaboration of TECNALIA RESEARCH & INNOVATIONS in the implementation of the CPS and for their expertise in electronics and communication.

**Conflicts of Interest:** The authors declare no conflict of interest.

## Glossary

| | |
|---|---|
| $C_{AL}$ | Cost of availability losses |
| $C_{LI}$ | Cost Loss Indicator |
| $C_{PL}$ | Cost of performance losses |
| $C_{QL}$ | Cost of quality losses |
| $C_{QL_{rej}}$ | Cost of quality losses due to rejects |
| $C_{QL_{rew}}$ | Cost of quality losses due to reworks |
| $MC_{rej}$ | Material cost due to rejects |
| $OC_A$ | Opportunity cost of availability |
| $OC_P$ | Opportunity cost of performance |
| $OC_{Qrej}$ | Opportunity cost of quality due to rejects |
| $PC_A$ | Production cost of availability |
| $PC_P$ | Production cost of performance |
| $PC_{Qrej}$ | Production cost of quality due to rejects |
| $RC$ | Rework Cost |

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
