# Peer review of "Development of a New KPI for the Economic Quantification of Six Big Losses and Its Implementation in a Cyber Physical System"

_applsci, doi:10.3390/app10249154_

Round 1

Reviewer 1 Report

The article presents the development of a new KPI which measures the 6 big losses in an economic dimension and can be implemented in a CPS to its real-time calculation. It supports the idea that Industry 4.0 enhances the machining tool analysis and the decision-making for companies. It is a very interesting and new topic that has the potential for publishing but I have some recommendations for its improvement:

In the abstract of the paper, it is necessary to clearly emphasize the aim of the paper, research methodology, and main obtained results.

The introduction should contain:

  • the main purpose of the paper,
  • an explanation - why is the subject of the paper significant for the research?,
  • the hypothesis of the research,
  • an explanation of applied research methodology.

The article does not mention either when the survey was conducted, or in what company/companies.

On line 55, there is written that Lean methodology is receiving much attention in recent years and is one of the most 55 common strategies to improve operational capabilities. Lean methodology is older than for example Industry 4.0 and literature older than 10 years is also cited here (from 2009).

The objectives of the research are described at the end of Chapter 1. Specify these goals in more detail.

Other recommendations:

  • unify citations in the text,
  • several times in the article it is written: Error! Reference source not found.,
  • figure 4 is illegible,
  • indicate which companies/researchers the results are applicable to.

Author Response

Dear Reviewer,

Once all the reviewers’ comments have been considered, we are going to submit our revision manuscript and we would like to detail the revision of both reviewers point by point, therefore we have answered next to the reviewers’ comments. Our comments are wrote in blue at the pdf to distinguish them easily. Moreover, we have rephrased some parts of the paper to achieve a self-cited rate lower than 10% and we have added a Glossary at the end of the paper to enhance the equation comprehension.

REVIEWER 1

The article presents the development of a new KPI which measures the 6 big losses in an economic dimension and can be implemented in a CPS to its real-time calculation. It supports the idea that Industry 4.0 enhances the machining tool analysis and the decision-making for companies. It is a very interesting and new topic that has the potential for publishing but I have some recommendations for its improvement:

In the abstract of the paper, it is necessary to clearly emphasize the aim of the paper, research methodology, and main obtained results. It has been emphasized in lines 11, 13 and 20.

The introduction should contain:

  • the main purpose of the paper, Line 178 – 185. Therefore, the objective of this study is to fill those gaps developing a cost model and a new cost KPI capable of measuring in real-time the impact of cost in the six big losses developed by Nakajima. Both, the cost model and the KPI, are implemented in a CPS, which allows us their real-time calculation. Not only does this new KPI enhance the decision-making, but it also shows the most effective points to decrease production costs. Moreover, it completes the dimensions in which the 6 big losses can be represented, representing them in an economic dimension and allowing companies to develop a multidisciplinary decision-making tool, comparing the 6 big losses in a productive, sustainable and economic dimension.

  • an explanation - why is the subject of the paper significant for the research? We have further explained the gaps filled by our research, from line 173 to 177. It can be deduced that working in real-time has not received much attention yet, even though Industry 4.0 enhances its use, e.g., using CPS for real-time data acquisition. The use of this CPS allows the acquisition of time variables which are more accurate than the average times, that are often used as an approximation. Regarding to the six big losses research, it has been further developed in terms of productivity, but not in other dimensions, such as the sustainable or economic ones.

  • the hypothesis of the research. We have not a hypothesis of the research, but the hypothesis of our case of study are shown in lines 370 – 371.

  • an explanation of applied research methodology. Line 186-191. After the literature review of the key research concepts and their correlation. This manuscript goes ahead with the following research methodology. A cost model has been analysed and adapted to our research for developing our new KPI and its equations. After that it has been implemented in a CPS. On that note, a case of study has been carried out to show this KPI’s development and implementation. The results have been discussed and compared with other KPIs. The paper ends with conclusions and proposals for future studies.

The article does not mention either when the survey was conducted, or in what company/companies. Line 297 Our CPS is located in the Design and Manufacturing Engineering Department of the University of Zaragoza, where we are creating a Lab 4.0 in collaboration with Tecnalia.

On line 55, there is written that Lean methodology is receiving much attention in recent years and is one of the most 55 common strategies to improve operational capabilities. Lean methodology is older than for example Industry 4.0 and literature older than 10 years is also cited here (from 2009). This older reference has been changed. In lines 60 - 62 two new references of 2019 and 2020 have been added. Lean methodology framework is being an interesting topic for many researchers in recent years [14] and is one of the most common strategies to achieve better results and higher competitiveness [15].

  • The objectives of the research are described at the end of Chapter 1. Specify these goals in more detail. Lines 173 – 185. It can be deduced that working in real-time has not received much attention yet, even though Industry 4.0 enhances its use, e.g., using CPS for real-time data acquisition. The use of this CPS allows the acquisition of time variables which are more accurate than the average times, that are often used as an approximation. Regarding to the six big losses research, it has been further developed in terms of productivity, but not in other dimensions, such as the sustainable or economic ones. Therefore, the objective of this study is to fill those gaps developing a cost model and a new cost KPI capable of measuring in real-time the impact of cost in the six big losses developed by Nakajima. Both, the cost model and the KPI, are implemented in a CPS, which allows us their real-time calculation. Not only does this new KPI enhance the decision-making, but it also shows the most effective points to decrease production costs. Moreover, it completes the dimensions in which the 6 big losses can be represented, representing them in an economic dimension and allowing companies to develop a multidisciplinary decision-making tool, comparing the 6 big losses in a productive, sustainable and economic dimension.

Other recommendations:

  • unify citations in the text, citations have been unified.
  • several times in the article it is written: Error! Reference source not found., this problem has been solved.
  • figure 4 is illegible, figure 4 has been modified.
  • indicate which companies/researchers the results are applicable to. The majority of the companies, which are using Industry 4.0 technologies or are nearly to use them can implement this technology in several resources, i.e., machining tools, 3D printers or Automated Guide Vehicles (AGVs). (Line 440 – 442)

Sincerely,

Paula Morella

Reviewer 2 Report

Reading the paper, I was having difficulties to see what is its contribution in the field.

The type of costs analysed by the authors are not new and their mapping should be explained better.

For instance, the Figure 3 and the surrounding discussion should explain in more detail how the costs are mapped to KPI.

There is also a lot of similarity with the paper Development of a New Green Indicator and Its Implementation in a Cyber–Physical System for a Green Supply Chain.

The English has to be improved as well.

Cross-reference of the figures, tables and equations have to be fixed (a lot of errors in the document: "Error! Reference source not found").

Numeration of the figures is also incorrect.

Author Response

Dear Reviewer,

Once all the reviewers’ comments have been considered, we are going to submit our revision manuscript and we would like to detail the revision of both reviewers point by point, therefore we have answered next to the reviewers’ comments. Our comments are wrote in blue at the PDF to distinguish them easily. Moreover, we have rephrased some parts of the paper to achieve a self-cited rate lower than 10% and we have added a Glossary at the end of the paper to enhance the equation comprehension.

REVIEWER 2

Reading the paper, I was having difficulties to see what its contribution in the field is. The gaps which have been filled with our research appear in lines 173 – 185. It can be deduced that working in real-time has not received much attention yet, even though Industry 4.0 enhances its use, e.g., using CPS for real-time data acquisition. The use of this CPS allows the acquisition of time variables which are more accurate than the average times, that are often used as an approximation. Regarding to the six big losses research, it has been further developed in terms of productivity, but not in other dimensions, such as the sustainable or economic ones. Therefore, the objective of this study is to fill those gaps developing a cost model and a new cost KPI capable of measuring in real-time the impact of cost in the six big losses developed by Nakajima. Both, the cost model and the KPI, are implemented in a CPS, which allows us their real-time calculation. Not only does this new KPI enhance the decision-making, but it also shows the most effective points to decrease production costs. Moreover, it completes the dimensions in which the 6 big losses can be represented, representing them in an economic dimension and allowing companies to develop a multidisciplinary decision-making tool, comparing the 6 big losses in a productive, sustainable and economic dimension.

The type of costs analysed by the authors are not new and their mapping should be explained better. We are not looking for a new cost model, we are adapting a cost model which can be used to develop our KPI. As can be seen in line 197-198, The cost model presented by Lambán has been modified, focusing on direct costs, which are the most significance for our KPI development.

We added these lines to explain better the use of cost model in our research. This cost model considers the raw material and the different phases which are included in a product machining process, from its design to its manufacture. Below is an outline of our cost model (Line 209-212) The total sum of all cost model components gives as a result the cost per piece manufactured. However, we use the itemized cost model for our new KPI implementation, as can be seen in table 1 (Line 229-231).

For instance, the Figure 3 and the surrounding discussion should explain in more detail how the costs are mapped to KPI. We have extended our explanation. (Line 311 – 325) Once the CPS is implemented, the most appropriate variables for KPI calculation in real-time are analysed, checked, and selected. Python has been used to develop the software that translates the acquired data into relevant information and indicators. Our system can capture variables, such as machine time, energy consumption, part number, tool number, tool changes and the spindle rotation. However, other parameters which must be given, are stored in a database and can be seen in table 2. Then,  the software can calculate KPIs, such as OEE or ECL, which are defined in [28], our model cost defined in 2.1 and our new KPI defined in 2.2 using the acquired data and the established parameters. For example, to calculate the energy cost of six big losses, the CPS can capture the energy consumption associated with each one of the six big losses. As the CPS can determine the process that the system is carried out (how long it takes, how much energy the machine consumes, if the machine is working properly or not, and if the machine is working at reduced speed). Once the CPS knows this energy consumption, it can obtain from the database the energy cost per kWh. Figure 3 represents a flow chart for the KPIs implementation in a CPS, these KPIs will be compared later (OEE, ECL and ECI).

There is also a lot of similarity with the paper Development of a New Green Indicator and Its Implementation in a Cyber–Physical System for a Green Supply Chain. Regarding the similarities with our last paper: Development of a New Green Indicator and Its Implementation in a Cyber–Physical System for a Green Supply Chain. Both papers take part of our research in Six Big Losses KPIs development in different dimensions (sustainable, economic...) and its implementation in a Cyber Physical System (CPS). For that reason, our research methodology, the Six Big Losses definition, the case of study and the CPS implementation are similar in both cases, nevertheless the KPI development is completely different. Furthermore, these similarities allow us to compare our KPIs in the same scenery but in different dimensions, adding more value to our research. However, these similarities have been rephrased along the text.

The English has to be improved as well. English had been reviewed.

Cross-reference of the figures, tables and equations have to be fixed (a lot of errors in the document: "Error! Reference source not found"). This problem has been solved.

Numeration of the figures is also incorrect. We have not seen this error.

Sincerely,

Paula Morella

Round 2

Reviewer 2 Report

There are still errors in the text "Error! Reference source not found". When saving the file in Word (from docx to pdf) you should deactivate "update fields" options because it gives you errors.

The figure numbering problem is still here. For instance, you have figures 1-5 and then on page 12 you start again from 1 (it should continue, 6, 7 and so on).

Author Response

Dear Reviewer,

Thank you for all your comments.

Regarding to figures enumeration, we have not seen this error. As you can see in the Figure at the cover letter, we have figure 6 and not figure 1 again. We don’t now where is the problem; however, we have submitted the manuscript in PDF in case there is a problem with the Word versions.

We have solved the errors "Error! Reference source not found" too.

Sincerely,

Paula Morella
